# Acceptance Intention and Behavioral Response to Soil-Testing Formula Fertilization Technology: An Empirical Study of Agricultural Land in Shaanxi Province

**DOI:** 10.3390/ijerph20020951

**Published:** 2023-01-04

**Authors:** Hao Dong, Yang Zhang, Tianqing Chen, Juan Li

**Affiliations:** 1Institute of Land Engineering and Technology, Shaanxi Provincial Land Engineering Construction Group Co., Ltd., Xi’an 710075, China; 2School of Management, Xi’an Jiaotong University, Xi’an 710049, China

**Keywords:** acceptance intention, soil-testing formula fertilization technology, theory of planned behavior, PLS-SEM, well-facilitated farmland construction

## Abstract

Soil-testing formula fertilization technology is a powerful tool for preserving arable land and ensuring food security. The purpose of this study was to investigate farmers’ acceptance intentions and behavioral responses to soil-testing formula fertilization technology. Based on the theory of planned behavior, this paper adopts the partial least squares structural equation modeling (PLS-SEM) method, with 295 farmers in the high-standard farmland project area of Shaanxi Province as samples. The research results show that attitude (ATT), subjective norms (SN), and perceived behavioral control (PBC) all had a significant positive influence on farmers’ behavioral intentions. The order of impact effects from large to small is ATT > SN > PBC. The ecological rationality of farmers, communication and demonstration between neighbors, and effective technical training, as well as consulting and guidance services, can better enhance farmers’ intentions to apply soil-testing formula fertilization technology. This study could help to provide references for policymaking to improve the adoption of soil-testing formula fertilization technology.

## 1. Introduction

China is a main grain producer and feeds nearly 20% of the world’s population with 9% of the world’s arable land [1,2,3]. Fertilizer is used as a material to improve soil fertility all over the world, but the overuse of fertilizer causes serious non-point source pollution: this is recognized as a problem in the world [4]. The large-scale application of chemical fertilizer not only destroys the soil structure of cultivated land, but also accelerates the loss of nutrients and promotes the serious compaction of soil [5]. Moreover, it leads to the decline of the quality of agricultural products and the excessive content of nitrates in crops, which seriously threatens people’s health [6].

Farmers are the main body of agricultural production, and their fertilization behavior affects agricultural non-point source pollution [7,8]. Therefore, the key to the treatment of agricultural non-point source pollution is to encourage farmers to adopt environmentally friendly agricultural technologies, either voluntarily or through government incentives. As an environmentally friendly agricultural technology, the application of soil-testing formula fertilization (STFF) technology can effectively alleviate agricultural non-point source pollution [9,10]. Soil testing and fertilizer application technology is a method of precise fertilization based on soil nutrient test results and fertilizer field trials, and the growth elements required by crops [11]. It can effectively solve the problems existing in fertilizers. However, according to the Ministry of Agriculture and Rural Affairs, the adoption rate of STFF technology in China was not expected to reach 30 percent by the end of 2021 [12]. There are two main reasons for the low adoption rate. On the one hand, the impact of STFF technology on environmental performance is significant, but its effect cycle time is too long. On the other hand, mastering STFF technology requires farmers to invest time and shoulder the economic costs, and the need to bear risks is the main reason for the low adoption rate. Finally, there is “disconnection” or “deviation” between farmers’ intentions to participate and their adoption behavior.

In recent years, existing studies on farmers’ intentions and behaviors in adopting technology explain, to a certain extent, why the large-scale popularization and application of STFF technology were so successful, but there are still shortcomings and room for improvement [9]. First of all, current research on environment-friendly technologies mainly focused on adoption behavior [13]. However, under the premise that the environment-friendly STFF technology was not applied on a large scale, it is difficult to obtain relevant data for farmers’ technology-adoption behavior. Nevertheless, it is an effective way of analyzing their adoption intention [14]. Secondly, existing studies ignored the influence of psychological factors on farmers’ decision making in terms of the dimensions of influencing factors [15]. STFF technology is a government-led, business, and farmer participation mode, which not only requires promotion by external forces, but also the driving force of the farmers themselves [16]. Farmers’ perceptions of technology and trust in policies directly affect the effect of policy implementation [17]. Finally, existing studies mostly used Probit or Logit models to analyze influencing factors, and the selection of variables was relatively scattered [18,19]. There are few studies on the influence path of farmer adoption of STFF technology. In conclusion, this paper takes farmers in well-facilitated farmland construction projects in Shaanxi Province as the research objects, and uses the partial least squares structural equation model, based on planned behavior theory, to explore the adoption intention and behavioral response rules pertaining to farmers and STFF technology. This paper provides a reference for promoting change from farmers’ intentions to adopt environmentally friendly farmland protection technology to their behavioral response, and a change in their perception of policy formulation, to improve the quality of arable land and ecological protection in China.

The structure of the article is as follows: Section 2 summarizes the theory and hypotheses; Section 3 introduces the questionnaire and data source; Section 4 presents the results of the study. Section 5 summarizes the conclusions and contributions, and provides practical implications based on empirical findings.

## 2. Theory and Hypotheses

### 2.1. Theory of Planned Behavior

The theory of planned behavior (TPB) is often used to study the influence of behavioral intention (BI) on individual behavior [20,21]. The theory of planned behavior involves three key factors: attitude (ATT), subjective norms (SN), and perceived behavior control (PBC) [22]. The variables at the three levels are relatively independent but interact with each other at the same time [23]. The theory was widely used to predict and explain the correlation of individual cognition, intention, motivation, and behavioral decision making, and achieved remarkable results [24]. According to the technical characteristics of STFF, combined with the relevant requirements of planned behavior theory, this paper explores the transformation process of “cognition-intention-behavior” of farmer use of STFF from three dimensions of individual cognitive setting questions. Figure 1 shows the conceptual framework.

### 2.2. Attitude (ATT)

ATT refers to an individual’s positive or negative feelings towards specific behaviors [25]; this can be understood as a farmer’s expectations of the advantages or disadvantages of STFF technology. Farmers are not only “rational economic man”, but also “ecological economic man” [25,26]. On the one hand, farmers possess the innate characteristics of paying attention to agricultural production benefits and pursuing the maximization of economic benefits [27]. Their expectation of STFF technology in reducing production costs and increasing benefits is an important factor affecting their behavior judgment [28]. On the other hand, while pursuing economic profits, farmers will also evaluate the role and value of STFF technology in ecological protection according to their own understanding of cultivated land and ecological environment protection [29]. The more positive the expectation of the ecological protection function of STFF technology, the stronger the intention to adopt it. In line with some of the abovementioned insights, this reasoning leads to the following hypothesis:

**Hypothesis** **1** **(H1).**
*ATT has a positive effect on farmer BI of using STFF technology.*


### 2.3. Subjective Norms (SN)

The term SN refers to the influence of social pressure on individual behavioral decision making [30]. It can be understood as the guiding and demonstration role of individuals or groups that are closely related to the farmers in their decision making in terms of using STFF technology. Wang and Tan proposed that SN can be further divided into demonstrative norms and prescriptive norms [31]. Demonstrative norms refer to the driving effect of behavior conduction and production communication among relatives and neighbors in the adoption of STFF technology by farmers [32], while prescriptive norms refer to the extent to which the local government promotes the STFF technical information and the extent to which village cadres support the technology [33]. In addition, whether the government promotes the technology widely and provides strong support, or communication among neighbors promotes a good demonstration effect [34], both have a significant promoting effect on the behavior and ATT of farmers. The behavior intention of farmer use of STFF is thus impacted through the “bridge” of behavior and ATT. Based on the above analysis, this paper takes two aspects of demonstrative norms and prescriptive norms into consideration, and selects indicators from two levels of neighborhood demonstrative norms and prescriptive norms. According to this line of reasoning, we propose the following hypotheses.

**Hypothesis** **2** **(H2).**
*SN have a positive effect on ATT of STFF technology.*


**Hypothesis** **3** **(H3).**
*SN have a positive effect on BI of STFF technology.*


### 2.4. Perceived Behavior Control (PBC)

PBC refers to an individual’s judgment about the difficulty of completing a certain task [35]; it is specifically manifested as an individual’s judgment of the resources and opportunities they have mastered and their operational ability. In practice, individuals are often unable to actually engage in a certain behavior due to the lack of ability to control resources [36]. PBC can be further elaborated from two angles: self-efficacy and external environment analysis [37]; self-efficacy refers to an individual’s ability to undertake specific behaviors at various levels of competence [38], and external resources refers to an individual’s environment that provides the opportunities and conditions for engaging in a particular behavior [38]. PBC influences individual behavior in two ways: first, PBC has a significant promoting effect on the generation of BI [39]; second, PBC reflects, to some extent, the completeness of the actual control conditions the farmer experiences [40], which will directly affect their technology-adoption behavior. In terms of PBC for STFF, the self-efficacy of farmers can be understood as the self-mastery ability of using STFF technology and the acquisition ability of formula fertilizer. For farmers, the external environment can be interpreted as the relevant professional and technical personnel who provide the corresponding consultation and guidance services, and effective training in the use of STFF technology. The stronger a farmer’s ability to master and acquire STFF technology, the more effective the consultation and guidance services, and training, provided by relevant professional and technical personnel, and the stronger the farmer’s intention and application behavior in adopting the technology itself. Based on this, this paper measures farmers’ PBC from two aspects: self-efficacy and external environment. Consequently, we propose the following hypotheses:

**Hypothesis** **4** **(H4).**
*PBC has a positive effect on BI in implementing STFF technology.*


**Hypothesis** **5** **(H5).**
*PBC has a positive effect on technology application behavior (TAB).*


In addition, according to the theory of planned behavior, when the actual control conditions are fully satisfied, all factors affecting the behavior act on the behavior itself through influencing BI. Consequently, we propose the following hypothesis:

**Hypothesis** **6** **(H6).**
*BI has a positive effect on TAB.*


## 3. Methodology

### 3.1. Questionnaire Design

Based on the theory of planned behavior, this paper includes five latent variables: behavioral ATT, SN, PBC, BI, and STFF behavior. The observed variables of each latent variable were measured using a 7-point Likert scale (1 = strongly agree, 7 = strongly disagree). STFF-adoption behavior was measured by three dimensions: whether to use it, the proportion of use, and the number of years of use. The proportion use is the ratio of the applied area of STFF technology to the actual cultivated area, and the number of years of use is the number of years that farmers had been using STFF technology, as presented in Appendix A.

### 3.2. Data Collection

All the data used in this paper are from the research team’s survey on Well-facilitated Farmland Construction Projects in Shaanxi Province, from April to September 2022; the sample areas were randomly stratified in Xi’an, Weinan, Baoji, Yulin, and Shangluo in Shaanxi Province. Eight towns and 16 villages were randomly selected for the household survey. A total of 400 questionnaires were distributed in this survey, with 295 valid samples being returned (an effective rate of 73.75%). The contents of the questionnaire included individual characteristics of farmers, family characteristics, tillage protection and cognition of STFF technology, adoption of STFF technology, and measurement of behavioral ATT, SN, and PBC.

### 3.3. Data Analysis

PLS-SEM was conducted using the SmartPLS Vision 3.3.2 Software Tool to analyze and interpret the model. The partial least squares structural equation model (PLS-SEM) was used to model the relationship between latent variables [41]; this was effective in solving the difficult issues of directly observing farmer cognition and clearly describing the decision-making process in farmer behavior; it also allowed analysis of the influence of variables and whether there were an differences [42]. We divided the analytical process into two parts according to Hair’s suggestion: firstly, the latent construct dimensions, validity and reliability of the measurement model were evaluated. Secondly, the path coefficient and path significance of the structural model were evaluated. The PLS algorithm was used to derive the path coefficients of the structural model, and the path weighting-scheme algorithm was used to give the standardized regression coefficients [43]. The statistical significance of the structural path was evaluated by the bootstrapping procedure.

### 3.4. Sample Characteristics

As shown in Table 1, 295 valid questionnaires were collected. The characteristics of the samples were as follows: 168 (56.9%) males and 205 (69.5%) middle-aged adults over 50 years old. Of these, 225 (76.3 percent) were not educated to high school level, accounting for more than three-quarters of the sample size. Only 21 of the subjects were village cadres. The results of the mean and variance of latent variables are shown in Table 2: ATT (M = 5.000, SD = 1.082), SN (M = 4.714, SD = 1.116), PBC (M = 4.894, SD = 1.148), BI (M = 5.223, SD = 1.176), and TAB (M = 5.066, SD = 1.082).

### 3.5. Common Method Variance

To address the problem of common method variance, it was emphasized during data collection that there was no need to sign the question paper and there were no right or wrong answers. The information provided by participants in the questionnaire was only to be used for academic research and would be kept strictly confidential. The Harman single-factor test was used for homologous analysis of variance; unrotated principal component analysis of all items of the five study variables showed that the proportion of unrotated first factor in all explanatory variables was 39.65%, much lower than the threshold of 50%. The method factor was then introduced to the test and a two-factor model was established; that is, a common method factor was added to the structural equation model as a global variable, and the fitting degree changes of the skew factor model and the two-factor model were compared. The results show that after controlling the common method factors, the model fit degree did not change significantly (ΔX^2^ = 14.58, Δ*df* = 9), so the homologous variance problem was not serious.

### 3.6. Reliability and Validity

In this study, Mplus7.4 and SmartPLS3.0 were used to test the reliability and validity of variables. First, Mplus7.4 was used for the confirmatory factor analysis of key variables, and the results show that the five-factor model had a good fitting effect (X^2^ = 592.511, *df* = 242, *p* < 0.01; SRMR = 0.047, RMSEA = 0.055, CFI = 0.915, TLI = 0.903); the goodness of fit of the five-factor model was significantly better than that of the other four models (see Table 3), indicating that the concept of the five factors is clear and can be effectively distinguished. Therefore, it was appropriate to construct the model with five factors. SmartPLS was used for reliability and validity analyses (see Table 4). The load values of all construct factors ranged from 0.735 to 0.932. At a significance level of 0.001, Cronbach’s α ranged from 0.776 to 0.888. Combination reliability ranged from 0.856 to 0.930. The results show that all variables have high internal consistency and combination reliability [42,44]. The extraction variance (AVE) of all constructs was greater than the threshold value of 0.5, indicating that the model has good convergent validity. The square roots of all variables AVE were greater than the correlation coefficients between this construct and other constructs, indicating good discriminative validity of the model (see Table 5). The Heterotrait–Monotrait ratio was used to evaluate the zone validity. This method was more sensitive to the variance-based structural equation validity problems [45]. The ratio was found to be below the 0.85 threshold (see Table 5). In conclusion, the measurement model met the basic requirements of reliability and validity. To avoid issues associated with CMB, determining the variance inflation factor (VIF) and confirming collinearity are required [46]. As shown in Table 1, the VIF values of latent variables in the measurement model were all less than the threshold value 5, indicating that there is no multicollinearity between the measurement variables.

## 4. Results

### 4.1. Path Analysis

The model analysis results were obtained using SmartPLS 3.0, as shown in Figure 2 and Table 6. The results of path analysis show that ATT, SN and PBC had positive effects on farmer BI (β = 0.245, *p* < 0.001; β = 0.221, *p* < 0.001; β = 0.201, *p* < 0.001), H1, H3, and H4 were all verified. SN have a positive impact on ATT (β = 0.679, *p* < 0.001), PBC has a positive impact on TAB (β = 0.389, *p* < 0.001), and farmer BI has a positive impact on TAB (β = 0.450, *p* < 0.001). H2, H5, and H6 were all verified. The R^2^ of BI suggests that ATT, SN, and PBC explain the 32.1% effect of STFF technology adoption intention. The R^2^ of TAB suggests that PBC and BI explain the 41.1% effect of STFF TAB. It can be seen that the explanatory power of the model is strong. By observing Q^2^ of endogenous variables to evaluate the predictive ability of the model, the results show that ATT (Q^2^ = 0.313), BI (Q^2^ = 0.256) and TAB (Q^2^ = 0.326) are all greater than 0, and the model has good predictive correlation.

### 4.2. Mediating Effect

Behavior Intention was checked through the bootstrapping program and confidence intervals. The bootstrapping resampling was set to 5000 times, and the results show (see Table 6) that BI plays a significant role in mediating among ATT, SN, PBC, and TAB. H2a and H2b were verified, indicating that resource management plays a key role in enterprise ecological innovation. At the same time, it is proven that compared with internal resources, the direct, indirect, and total effects of external resources on enterprise ecological innovation are more significant. The standard of the test was to see whether the confidence interval contains 0. As shown in Table 6 and Figure 2, all hypotheses for the three mediation effects were verified.

## 5. Discussion

In China, which has a large population but has far less arable land per capita than the rest of the world, strengthening cultivated land protection, realizing sustainable use of cultivated land, and ensuring food security are critical issues for the national economy and people’s livelihood. Data from the National Bureau of Statistics show that before 2015, in China, the trend of using fertilizer was increasing annually. In order to curb the excessive use of chemical fertilizers and ensure the quality and safety of agricultural products, and the sustainable development of the ecological environment, China’s Ministry of Agriculture issued the National Action Plan for the Popularization of Soil-Testing Formula Fertilization Technology in 2012; this was a clear proposal to accelerate the transformation of chemical fertilizer use and vigorously promote green farmland protection technology. Although policies to reduce fertilizer use have yielded significant benefits over the past decade, farmers were not active in adopting STFF technology, with only a third of farmers choosing to apply the technology, according to the Ministry of Agriculture, in 2021. This may be due to the fact that STFF technology is a new alternative technology for protecting arable land and requires farmers to have the relevant knowledge and operational skills. However, the promotion of STFF technology is difficult. In terms of arable land, the Guanzhong plain area in Shaanxi includes a wide area of arable land and good soil fertility, but the quality of arable land in areas such as the Loess Plateau and the mountains of southern Shaanxi is poor, making it difficult to use STFF technology effectively. Due to the limited availability of information, farmers cannot keep themselves informed of the outside world’s environmental protection concepts and new agricultural technologies. STFF technology is within the limits of the ecological carrying capacity and environmental capacity, and will not cause harm to the cultivated land ecosystem. It is beneficial for the protection of the cultivated land ecological environment and can meet the needs of social production.

This study contributes to the diffusion of arable land conservation technology in China. At the same time, it fills the gap in terms of the factors influencing farmers’ acceptance of technology. This paper examines Chinese farmers’ intentions and behaviors regarding their use of STFF technology according to the theory of planned behavior, and obtains some useful findings. The factor load of ecological rationality was greater than that of economic rationality [47], indicating that the ecological rationality of farmers could promote the generation of their STFF behavior intention more than economic rationality; this may be related to the improvement in farmers’ living standards and the continuous improvement in ecological environmental protection awareness. This also shows that farmers, as direct users of cultivated land, are also the implementers of the concept of green development, and that ecological value plays an extremely important role in their cognitive system, gradually transforming from “rational economic man” to “ecological economic man”. Neighborhood communications between farmers, the exemplary role demonstrated in government propaganda and technology promotion [48], and policy incentives to stimulate farmer intentions to use soil testing and fertilizer application accord with other scholars’ research results, such as those of Liu et al. [49]. This may be because the face of a new technology, and the behavior and views of other farmers nearby, greatly affect their judgment; consequently, they then display herding behavior. The effective implementation of soil testing and fertilizer application technology not only requires farmers to overcome their own fear of difficulties, but also requires that professional and technical personnel be equipped to provide consultation and guidance services in each link, as well as timely answers to farmers’ questions; this would enhance farmer confidence in mastering the technology.

## 6. Conclusions

First, the ATT, SN, and PBC of farmers all significantly promoted the BI of using STFF technology, and the order of effects was as follows: Behavioral ATT (0.388) > SN (0.245) > PBC (0.216). The promoting effect of PBC on TAB can be indirectly influenced by the influence of BI.

Second, the economic rationality of ATT can promote the application intention of using STFF technology more than the ecological rationality. The communication and demonstration between the neighbors of farmers can stimulate STFF technology application intention more than the publicity, technology promotion, and policy incentives by the government. Timely and effective technical training, and consultation and guidance services, can better stimulate the application intention of using STFF technology more than farmer self-mastery of technology.

Third, the farmer PBC of STFF technology significantly promoted their “intention-behavior” transformation. The application of STFF technology was significantly influenced by BI.

The data collection was limited by the region, but it can provide some policy suggestions for decision makers, and subsequently promote STFF technology among groups with “fragmented arable land and small per capita scale” in China. In addition, our research team will undertake further research in the following aspects. First, this study is one of the few applying STFF technology to cultivated land protection. Previous studies mainly focused on resource endowment, policy environment, and technical feasibility. This study focuses on the subjective behavior ATT in farmers. The next step could be to further develop the area by integrating TAM and TPB. Second, the characteristics of farmers in this study are regional. For example, farmers in western China depend more on water than fertilizer due to climate drought, so future studies could be conducted to expand the population distribution area.

## Figures and Tables

**Figure 1 ijerph-20-00951-f001:**
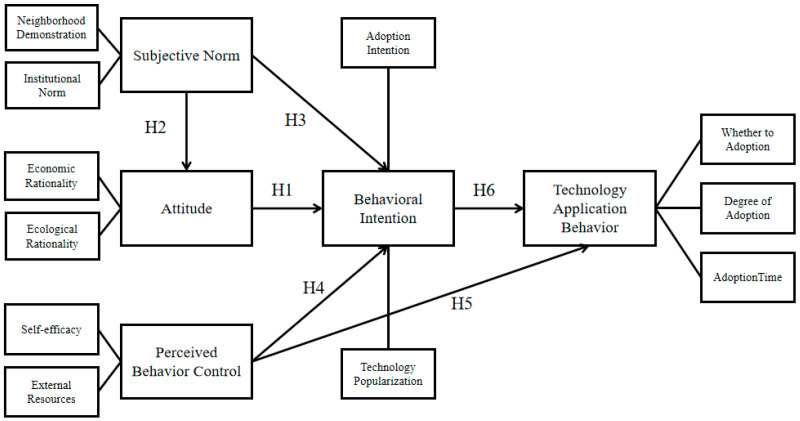
Research Model.

**Figure 2 ijerph-20-00951-f002:**
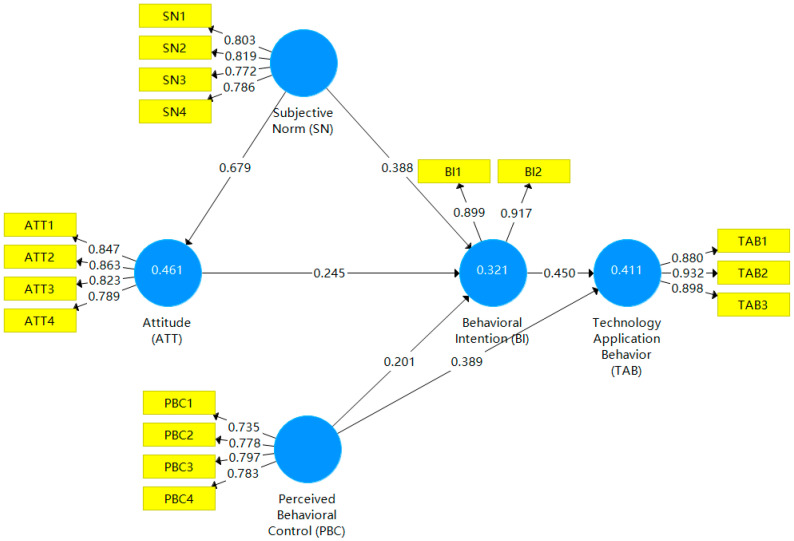
Path diagram of modified model.

**Table 1 ijerph-20-00951-t001:** Sample characteristics.

Variables	Definitions	Frequency	Proportion
Gender	Female	127	43.1%
	Male	168	56.9%
Age	20–30 years old	15	5.1%
	31–40 years old	24	8.1%
	41–50 years old	51	17.3%
	51–60 years old	185	62.7%
	61 years old and above	20	6.8%
Education	Not been to school	45	15.3%
	Primary school	122	41.4%
	Middle school	58	19.7%
	high school	80	27.1%
	College	10	3.4%
Cadre	Cadre	21	7.1%
	Non-cadre	274	92.9%

**Table 2 ijerph-20-00951-t002:** Descriptive statistics.

Variables	No. Items	Mean	SD
Attitude	4	5.000	1.082
Subjective Norms	4	4.714	1.116
Perceived Behavior Control	4	4.894	1.148
Behavioral Intention	2	5.223	1.176
Technology Application Behavior	3	5.066	1.082

Note: SD represents the standard deviation.

**Table 3 ijerph-20-00951-t003:** Confirmatory factor analysis results.

Model	X^2^	df	CFI	TLI	SRMR
Five factor model	592.511	242	0.915	0.903	0.047
Four factor model	725.438	246	0.884	0.870	0.052
Three factor model	796.528	249	0.867	0.853	0.055
Two factor model	859.382	251	0.853	0.838	0.056
One factor model	882.621	252	0.847	0.833	0.057

Note: Four-factor model: ATT and SN; three-factor model: ATT, SN, and PBC; two-factor model: ATT, SN, PBC, and BI; one-factor model: all variables were combined into one factor.

**Table 4 ijerph-20-00951-t004:** Reliability and validity.

Variables	Items	Loadings	Cronbach’s α	CR	AVE	VIF
Attitude	ATT1	0.847	0.850	0.899	0.690	2.071
	ATT2	0.863				2.362
	ATT3	0.823				2.024
	ATT4	0.789				1.633
Subjective Norms	SN1	0.803	0.806	0.873	0.632	1.624
	SN2	0.819				1.734
	SN3	0.772				1.562
	SN4	0.786				1.628
Perceived Behavior Control	PBC1	0.735	0.776	0.856	0.599	1.395
	PBC2	0.778				1.558
	PBC3	0.797				1.605
	PBC4	0.783				1.559
Behavioral Intention	BI1	0.899	0.788	0.904	0.825	1.733
	BI2	0.917				1.733
Technology Application Behavior	TAB1	0.880	0.888	0.930	0.817	2.123
	TAB2	0.932				3.628
	TAB3	0.898				2.980

Note: CR: Composite Reliability; AVE: Average Variance Extracted; VIF: Variance Inflation Factors.

**Table 5 ijerph-20-00951-t005:** Discriminant validity—Fornell–Larcker Criterion and Heterotrait–Monotrait Ratio.

Variables	ATT	BI	PBC	SN	TAB
Attitude (ATT)	**0.831**	0.610	0.650	0.820	0.745
Behavioral Intention (BI)	0.502 **	**0.908**	0.568	0.615	0.691
Perceived Behavior Control (PBC)	0.529 **	0.445 **	**0.774**	0.651	0.600
Subjective Norm (SN)	0.679 **	0.492 **	0.517 **	**0.795**	0.708
Technology Application Behavior (TAB)	0.648 **	0.583 **	0.498 **	0.598 **	**0.904**

Note: ** Correlation is significant at the 0.01 level (2-tailed), Bold diagonal entries are square root of AVEs, Heterotrait–Montrait ratios (HTMT) (Underlined) are below 0.85.

**Table 6 ijerph-20-00951-t006:** Results of hypothesis testing.

Hypothesis	Effect	Path	Path Coefficient	Lower (2.5%)	Upper (97.5%)	t-Statistics	*p*-Value	Decision
Direct Relationships								
H1	Direct	ATT -> BI	0.245	0.101	0.384	3.402	0.001 ***	Accept
H2	Direct	SN -> ATT	0.679	0601	0.750	17.522	0.001 ***	Accept
H3	Direct	SN -> BI	0.221	0.049	0.377	2.680	0.001 ***	Accept
H4	Direct	PBC -> BI	0.201	0.087	0.322	3.316	0.001 ***	Accept
H5	Direct	PBC -> TAB	0.298	0.191	0.411	5.249	0.001 ***	Accept
H6	Direct	BI -> TAB	0.450	0.331	0.560	7.731	0.001 ***	Accept
Mediating Relationships								
H2*H6	Indirect	SN -> BI -> TAB	0.099	0.02	0.179	2.475	0.013 **	Accept
H2*H1	Indirect	SN -> ATT -> BI	0.167	0.068	0.269	3.266	0.001 ***	Accept
H2*H1*H6	Indirect	SN -> ATT -> BI -> TAB	0.075	0.026	0.133	2.714	0.007 **	Accept
H1*H6	Indirect	ATT ->BI -> TAB	0.110	0.040	0.191	2.864	0.004 **	Accept
H4*H6	Indirect	PBC -> BI -> TAB	0.090	0.039	0.148	3.212	0.001 ***	Accept
SRMR composite model = 0.047 R^2^_ATT_ = 0.461; Q^2^_ATT_ = 0.313 R^2^_BI_ = 0.321; Q^2^_BI_ = 0.256 R^2^_TAB_ = 0.411; Q^2^_TAB_ = 0.326								

Note: Significant level: *p* < 0.10; * *p* < 0.05; ** *p* < 0.01; *** *p* < 0.001.

## Data Availability

The data presented in this study are available within the article.

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
