# Peer review of "Acceptance Intention and Behavioral Response to Soil-Testing Formula Fertilization Technology: An Empirical Study of Agricultural Land in Shaanxi Province"

_ijerph, 2023, doi:10.3390/ijerph20020951_

Round 1

Reviewer 1 Report

General comments:

The manuscript “Acceptance intention and behavioral response of soil testing formula fertilization technology: An empirical study based on well-facilitated farmland construction projects in Shaanxi Province” investigated the acceptance and behavior of farmers in response to the adoption of the soil testing formula fertilizers (STFF) at a regional scale. The research topic is relatively novel and would be of interest to the general audience at IJERPH. However, several improvements are needed to make the manuscript more accessible and informative. My suggestion is moderate revision based on the following suggestions:

1. The writing of the manuscript needs to be extensively improved. Currently some of the sentences tend to be long and vague. The use of some terminologies is also questionable.

2. The discussion section should be expanded to inform the audience better about the implications of this work. For example, how would the adoption of STFF improve soil health and environmental quality? How to incentivize farmers to adopt STFF based on the findings of this work?

3. The authors might consider moving some of the detailed statistical analysis results to Appendix while only keeping the most relevant ones in the main texts. Currently the results section is too long.

4. A more thorough introduction is needed about STFF. From the abstract, it seems that STFF is a form of organic fertilizer. However, it was not stated clearly what STFF actually represents and how it is related to soil tests/ soil health.

5. The choice of the statistical approach was not fully justified. I also suggest including parameters used for building models (e.g. PLS).

Detailed comments:

Title: The title might be too long.

L11-14: Long sentences like this are slightly awkward and should be rewritten.

L15. ‘Micro effect’ seems like a vague term without a definition.

L43. Please state more here about what STFF is composed of.

L43-47. Awkward long sentence.

L65. ‘top-down systematic engineering’ is vague and hard to understand without definition.

L69. State why PLS is a superior choice to previous work using Probit or Logit models.

Figure 1. Consider adding abbreviations to the terms so readers can follow the corresponding sections easily with the help of this figure.

L191. Please state the parameters and software used to build the PLS algorithm and Bootstrapping procedure.

Figure 2. Please add abbreviations to the legend. Also, my suggestion is to focus the results section on the discussion of statistics derived from this figure, because it seems to be the core finding of this research.

Table 1. Is ‘Cadre’ a proper word here?

L210, L221-222. Such texts belong to the method section.

L277. ‘Land is small’ is not that accurate of a statement for China. Consider revising the terms to be more precise.

L310. Consider expanding the discussion section to talk more about the implication of this work. Are the results only applicable to Shanxi province? Can we extrapolate these conclusions to a broader scale? Are there gaps between the adoption of STFF and improving soil/ environmental health? What would be the future work?

L331. Replace the term “my research conclusion”. Also, consider shortening and combining paragraphs in the conclusion section.

Author Response

Response to Reviewer 1 Comments

On behalf of my co-authors, we thank you very much for giving us an opportunity to revise our manuscript, we appreciate editor and reviewers very much for their positive and constructive entitled “Acceptance intention and behavioral response to soil testing formula fertilization technology: An empirical study of agricultural land in Shaanxi Province”.

We have studied reviewer’s comments carefully and have made revision which marked in red in the paper. We have tried our best to revise our manuscript according to comments. Attached please find the revised version, which we would like to submit for your kind consideration.

We would like to express our great appreciation to you and reviewers for comments on our paper.

Looking forward to hearing from you.

Thank you and best regards.

Point 1: Title: The title might be too long.

Response 1: According to reviewer 3's suggestion, the title is: Acceptance intention and behavioral response to soil testing formula fertilization technology: An empirical study of agricultural land in Shaanxi Province.

Point 2: L11-14: Long sentences like this are slightly awkward and should be rewritten.

Response 2: Let's change the sentence "The purpose of this study was to investigate the effects of micro effects on farmers' intention to use soil testing formula fertilization technology." to "The purpose of this study was to investigate farmers' acceptance intentions and behavioral responses to soil testing formula fertilization technology.". What the original paper wanted to convey was that the study looked at the micro-behavior of farmer behavior rather than the macro and micro dimensions of policy and business.

Point 3: L43. Please state more here about what STFF is composed of.

Response 3: Soil testing formula fertilization technology (STFF) is a method of precise fertilization based on soil nutrient test results and fertilizer field trials, and according to the growth elements required by crops.

Point 4: L43-47. Awkward long sentence.

Response 4: Let's change the sentence "Let's change the sentence "The purpose of this study was to investigate the effects of micro effects on farmers' intention to use soil testing formula fertilization technology." to "The purpose of this study was to investigate farmers' acceptance intentions and behavioral responses to soil testing formula fertilization technology.". What the original paper wanted to convey was that the study looked at the micro-behavior of farmer behavior rather than the macro and micro dimensions of policy and business." to "Soil testing formula fertilization technology (STFF) is a method of precise fertilization based on soil nutrient test results and fertilizer field trials, and according to the growth elements required by crops.".

Point 5: L65. ‘top-down systematic engineering’ is vague and hard to understand without definition.

Response 5: Let's change the sentence "STFF technology is a multi-participation and top-down systematic engineering" to "STFF technology is a government-led, business and farmer participation mode”

Point 6: L69. State why PLS is a superior choice to previous work using Probit or Logit models.

Response 6: PLS-SEM can deal with SEMs that contains multiple causal relationships among one or

more latent variables and estimate the relationships without imposing distributional assumptions, especially in a context where the sample size is small (Ringle et al., 2012).  Furthermore, studies that examine the mediating effects rarely address the consequences of measurement error. To offset some of the effects of measurement error, PLS-SEM can be regarded as a second best option (Aguinis et al., 2016; Hair et al., 2017). Additionally, it is more suitable to develop and examine theory (Hair et al., 2017). Hence, PLS-SEM is conducted by using SmartPLS Vision 3.3.2 Software Tool to analyze and interpret the model.

Point 7: Figure 1. Consider adding abbreviations to the terms so readers can follow the corresponding sections easily with the help of this figure.

Response 7: We have added abbreviations to the change Figure 1.

Point 8: Please state the parameters and software used to build the PLS algorithm and Bootstrapping procedure.

Response 8: PLS-SEM is conducted by using SmartPLS Vision 3.3.2 Software Tool to analyze and interpret the model, and it requires two stages: analysis of measurement model and analysis of structural model.

Point 9: Figure 2. Please add abbreviations to the legend. Also, my suggestion is to focus the results section on the discussion of statistics derived from this figure, because it seems to be the core finding of this research.

Response 9: Our revised manuscript has partially adjusted the results and discussion sections.

Point 10: Table 1. Is ‘Cadre’ a proper word here?

Response 10: In the Chinese context, the influence of cadres on farmers is very significant. The word "cadre" in the questionnaire sample features is referred to Effects of Risk Perception of Pests and Diseases on Tea Farmers Green Control Techniques Adoption published in International Journal of Environmental Research and Public Health.

Point 11: L210, L221-222. Such texts belong to the method section.

Response 11: We adjusted the section you mentioned to the method section.

Point 12: L277. ‘Land is small’ is not that accurate of a statement for China. Consider revising the terms to be more precise.

Response 12: In China, which has a large population but whose per capita arable land area is far below the world average, strengthening the protection of cultivated land, realizing sustainable use of cultivated land and ensuring food security are urgent matters concerning the national economy and people's livelihood.

Point 13: L310. Consider expanding the discussion section to talk more about the implication of this work. Are the results only applicable to Shanxi province? Can we extrapolate these conclusions to a broader scale? Are there gaps between the adoption of STFF and improving soil/ environmental health? What would be the future work?

Response 13: We have rewritten the discussion section based on your suggestions in the revised manuscript.

Point 14: L331. Replace the term “my research conclusion”. Also, consider shortening and combining paragraphs in the conclusion section.

Response 14: The data collection was limited by the region, but it can provide some policy suggestions for decision-makers, and then promote the STFF technology for the groups with "fragmented arable land and small per capita scale" in China. Also, we have made a shortened revision of the conclusions.

Reviewer 2 Report

Dear Authors,

Detailed notes on the manuscript are given below:

1) Consider the possibility of shortening the title of the work (e.g. "Acceptance intention and behavioral response to soil testing 2 mixture fertilization technology - an empirical study of agricultural land in Shaanxi Province") - other information can be included in the "scope of work" chapter; to the Authors' decision

2) The content in lines 78-81 is redundant

3) If you want to include conclusions in the abstract, do not use bullet points.

4) You have formulated hypotheses 1-6, what about H0?

5) 3.3. Data Analysis - did you study the differences between the values ​​- was a significance level (alpha) adopted? In lines 217-219 you are referring to delta X after all

6) Tables 5 and 6 (Note: Significant level ....) - information should be included in the methodology (see point 5)

7) Chapter 6. "Conclusions" is a summary, I suggest changing the title to "Summary"

Author Response

Response to Reviewer 2 Comments

On behalf of my co-authors, we thank you very much for giving us an opportunity to revise our manuscript, we appreciate editor and reviewers very much for their positive and constructive entitled “Acceptance intention and behavioral response to soil testing formula fertilization technology: An empirical study of agricultural land in Shaanxi Province”.

We have studied reviewer’s comments carefully and have made revision which marked in red in the paper. We have tried our best to revise our manuscript according to comments. Attached please find the revised version, which we would like to submit for your kind consideration.

We would like to express our great appreciation to you and reviewers for comments on our paper.

Looking forward to hearing from you.

Thank you and best regards.

Point 1: Consider the possibility of shortening the title of the work (e.g. "Acceptance intention and behavioral response to soil testing 2 mixture fertilization technology - an empirical study of agricultural land in Shaanxi Province") - other information can be included in the "scope of work" chapter; to the Authors' decision

Response 1: According to reviewer 3's suggestion, the title is: Acceptance intention and behavioral response to soil testing formula fertilization technology: An empirical study of agricultural land in Shaanxi Province.

Point 2: The content in lines 78-81 is redundant.

Response 2: This part is important according to the journal style. The structure of the article is as follows: Section 2 summarizes the theory and hypotheses; Section 3 introduces the questionnaire and data source; Section 4 presents the results of the study. Section 5 summarises the conclusions, contributions, and provides some practical implications due to empirical findings.

Point 3: If you want to include conclusions in the abstract, do not use bullet points.

Response 3: The abstract delete the bullet points according to your suggestions.

Point 4: You have formulated hypotheses 1-6, what about H0?

Response 4: H0 is usually called the null hypothesis. And the corresponding H1-H6 holds the opposite hypothesis. For example, Hypothesis 1 (H1). ATT has a positive effect on farmers BI of STFF technology. Hypothesis 0 (H0). ATT has not a positive effect on farmers BI of STFF technology. Because H1 is verified, null hypothesis H0 is rejected.

Point 5: 3.3. Data Analysis - did you study the differences between the values ​​- was a significance level (alpha) adopted? In lines 217-219 you are referring to delta X after all

Response 5: We report the Chi-square, degrees of freedom, CFI, TLI, SRMR, and Cronbach’s α in Tables 3 and 4, respectively. Delta X is Delta c, means Chi-square. There was an error due to an editing problem.

Point 6: Tables 5 and 6 (Note: Significant level ....) - information should be included in the methodology (see point 5)

Response 6: We adjusted the section you mentioned to the method section. Table 5: **Correlation is significant at the 0.01 level (2-tailed). Significance was noted.

Point 7: Chapter 6. "Conclusions" is a summary, I suggest changing the title to "Summary"

Response 7: According to your suggestion, we have made a revision.

Reviewer 3 Report

This manuscript set out to discover the impact of micro effects on farmers' intentions to employ soil testing formula fertilization technologies. The manuscript is clearly written and well organized. Additionally, the study's goal merit is worth for  publication. This reviewer urges the authors to respond to the worries raised by this manuscript. Therefore, before publication, the manuscript needs a few small changes. Constructive comments for revision are:  

Minor comments:

Line 89: Please insert comma after motivation

Line 124: It would be more appreciable if you change the sentences “And then through the "bridge" of behavior ATT, the behavior intention of farmers' STFF was affected” by following ways

“Then the behavior intention of farmers' STFF was impacted through the "bridge" of behavior ATT”.

Line 135:  Please incorporate “they have” instead of he or she and their instead of his or her

Line 173: Please insert comma after Yulin

Line 247:  In table 4: It should be Cronbach’s α

 Line 254: Please insert comma after SN

Line 254: Please insert comma after H3

Line 269: Please insert comma after PBC

Line 277-279: This sentence is bit chaotic. Please fragment this sentence which would be easily perceivable to reader.

Author Response

Response to Reviewer 3 Comments

On behalf of my co-authors, we thank you very much for giving us an opportunity to revise our manuscript, we appreciate editor and reviewers very much for their positive and constructive entitled “Acceptance intention and behavioral response to soil testing formula fertilization technology: An empirical study of agricultural land in Shaanxi Province”.

We have studied reviewer’s comments carefully and have made revision which marked in red in the paper. We have tried our best to revise our manuscript according to comments. Attached please find the revised version, which we would like to submit for your kind consideration.

We would like to express our great appreciation to you and reviewers for comments on our paper.

Looking forward to hearing from you.

Thank you and best regards.

Point 1: Please insert comma after motivation

Response 1: The changes have been made in the revised manuscript.

Point 2: Line 124: It would be more appreciable if you change the sentences “And then through the "bridge" of behavior ATT, the behavior intention of farmers' STFF was affected” by following ways “Then the behavior intention of farmers' STFF was impacted through the "bridge" of behavior ATT”.

Response 2: The changes have been made in the revised manuscript.

Point 3: Line 135: Please incorporate “they have” instead of he or she and their instead of his or her

Response 3: The changes have been made in the revised manuscript.

Point 4: Line 173:Please insert comma after Yulin

Response 4: The changes have been made in the revised manuscript.

Point 5: Line 247:In table 4: It should be Cronbach’s α

Response 5: The changes have been made in the revised manuscript.

Point 6: Line 254:Please insert comma after SN

Response 6: The changes have been made in the revised manuscript.

Point 7: Line 254:Please insert comma after H3

Response 7: The changes have been made in the revised manuscript.

Point 8: Line 269:Please insert comma after PBC

Response 8: The changes have been made in the revised manuscript.

Point 9: Line 277-279: This sentence is bit chaotic. Please fragment this sentence which would be easily perceivable to reader.

Response 9: In China, which has a large population but has far less arable land per capita than the rest of the world, strengthening cultivated land protection, realizing sustainable use of cultivated land, and ensuring food security are critical issues for the national economy and people's livelihood.

Round 2

Reviewer 1 Report

The manuscript is improved after revision but can benefit from more thorough English editing.